# Post-Thaw Quality of Spermatozoa Frozen with Three Different Extenders in the Murciano Granadina Goat Breed

**DOI:** 10.3390/ani13020309

**Published:** 2023-01-16

**Authors:** Sonia Galián, Begoña Peinado, Laura Almela, Ángel Poto, Salvador Ruiz

**Affiliations:** 1Murcian Institute of Agricultural and Environmental Research and Development, Calle Mayor s/n, La Alberca, 30 150 Murcia, Spain; 2Department of Physiology, Faculty of Veterinary Medicine, University of Murcia, 30 071 Murcia, Spain

**Keywords:** goat, semen, freezing, skimmed milk, egg yolk, resistance

## Abstract

**Simple Summary:**

In caprine species, the semen quality of thawed samples still needs to be improved in order for this type of semen to provide an acceptable pregnancy rate when it is used for artificial insemination. It is therefore essential to have an adequate freezing protocol and extender for this species. By adapting freezing protocols that have proven successful in other species, such as pigs, together with the development of an extender that protects the sperm cells from damage caused by the freezing and thawing processes, and taking into account its composition in nutrients and cell-membrane-stabilizing substances, it has been possible, at least in vitro, to achieve high sperm quality in buck semen. In vivo studies of the fertilizing capacity of the semen cryopreserved in this way will be the next step to verify the effectiveness of the extender and semen-freezing protocol described in this work.

**Abstract:**

Artificial insemination (AI) with frozen semen in goats still presents serious difficulties, especially in certain goat breeds, in spite of technological progress. The aim of this work is the in vitro study of seminal extenders adapted from those used on other species to evaluate the response of goat sperm to several homeostatic conditions in order to achieve optimal post-thaw semen quality. Three different extenders based on different activity principles were used: (1) extender according to the methodology proposed for pigs, (2) skimmed-milk-based extender according to the methodology proposed for goats in France, and (3) a new egg-yolk-based extender replacing membrane-protective surfactants with sodium dodecyl sulfate (SDS) and named by our team as extender “IMIDA”. The freezing guidelines were those proposed for the freezing of porcine semen. The results obtained show that the egg-yolk-based extenders have good parameters of sperm motility at thawing, studied objectively using the computer-assisted sperm analysis (CASA) system and also subjectively. In particular, in the sperm resistance test after five hours, the thawed sperm containing SDS in their composition showed an optimal average on every evaluated parameter. The new IMIDA extender provided the highest sperm quality averages, so it could be a good extender to use in cryopreservation of semen in the caprine species.

## 1. Introduction

The dissemination of genetic characteristics of the males of interest in the goat species is greatly benefited if artificial insemination (AI) is performed with frozen semen. The use of frozen semen in AI programs offers the possibility of storing genetic material from valuable males for use over time, even after the death of the specimen [1], as well as its distribution to geographically distant farms without detriment to semen quality. However, in the caprine species, AI with frozen semen is still low due to the low fertility rate of the frozen semen. Whilst with fresh semen, the most common pregnancy rates are about 65–70% [2,3], sometimes reaching 80% [4], if frozen semen is used, they usually range from 20–40% [5,6].

The process of freezing–thawing semen is complex and has a number of critical points that destroy sperm cells, making the quantity and quality of sperm surviving it noticeably inferior to that of fresh semen, with a reduction of 30–40% in mobile thawed sperm compared to the percentage of mobile fresh sperm [7].

The use of a seminal extender is a determining factor for the quality of the semen after the freezing–thawing process and may make the difference between the success or failure of the technique. A good extender should provide the following:-Salts that maintain an appropriate osmotic pressure [8,9]. The extender must be isotonic with semen so as not to cause damage to the plasma membrane;-A buffer that cushions the changes in pH of the medium, neutralizing the acids produced by the sperm metabolism [10];-Sugars that serve as an energy source, amongst which fructose, glucose, trehalose, raffinose, ribose, sucrose, and galactose stand out. It should be taken into consideration that disaccharides do not penetrate into the cell, while monosaccharides do [9]. In addition, sugars also can improve cryoprotection by maintaining or increasing the osmotic pressure extracellularly in the sperm, which helps maintain the integrity of the membrane when stored for long periods of time;-Antibiotics to prevent the proliferation of microorganisms [11];-Cryoprotective agents that lower the eutectic point (minimum temperature at which the state of a solution is liquid) in order to prevent damage to the sperm plasma membrane as well as the formation of intracellular ice since they induce cell dehydration [8,12].

Generally, the semen freezing and refrigeration media of many species include egg yolk in a ratio of 15–20% [8]. The composition of egg yolk is a complex mixture of low-density lipoproteins (LDL) 68%, high-density lipoproteins (HDL) 16%, and 4% phosvitins [13]. Its sperm-preservation mechanism is not exactly known, but phospholipids, cholesterol, and LDL are believed to be the factors that provide sperm protection against cold shock by acting on the cell membrane, maintaining sperm motility, reducing the production of the acrosomal enzyme hyaluronidase, and maintaining mitochondrial membrane integrity [14,15]. Despite its advantages, the presence of egg yolk also has disadvantages, with the main one being the interaction that occurs between a phospholipase A, called EYCE enzyme (egg yolk coagulating enzyme) secreted by the bulbourethral glands, and lecithin in the egg, thereby producing lysolecithins and fatty acids. This would affect sperm quality, as reflected in reduced motility and viability, as well as decondensation of cryopreserved semen chromatin [16,17]. Removal of semen plasma by centrifugation avoids this problem [18].

Extenders based on skimmed milk have also been widely used. Milk components such as casein provide a protective effect [19], while other authors mention lactalbumin, which may act as buffer against pH changes and as a chelating agent of heavy metals [20]. It also presents the problem of interaction between a lipase enzyme secreted by the bulbourethral glands of goats, called BUSgp60, which interacts with triglycerides in skimmed milk and the sperm plasma membrane, producing an unsaturated fatty acid (oleic acid) that has a toxic effect on the sperm [21,22], resulting in a deterioration in motility and acrosome integrity and reduced in vitro sperm survival [16]. Sperm washing by centrifugation is therefore necessary when skimmed milk extenders are used.

Sodium dodecyl sulfate (SDS) is a detergent of the alkyl ionic group capable of denaturing the native protein structure. When added to freezing extender in small amounts, it has a beneficial effect on sperm motility and acrosomal integrity during freeze–thaw processes [23,24] and even prevents early sperm capacitation in frozen and thawed spermatozoa [25]. It has been suggested that the benefit that SDS provides on post-thaw sperm quality may be due to the modification it causes in the tertiary structure of the egg yolk lipoproteins of the extender [24,25], which exert their protective action on cell membranes [26]. The addition of SDS alone or containing commercial detergents such as Equex STM^®^ Paste (Orvus es Paste) to an egg-yolk-based extender improved sperm survival after thawing [25,27] and also the ability of sperm to bind to the zona pellucida of homologous oocytes [28,29]. It is therefore a common component in extenders for seminal freezing of different species.

The freezing protocol proposed by Thilmant (1997) [30] has been widely used by our team for years in porcine seminal cryopreservation. Due to the good results obtained in pigs, it would be interesting to check if this protocol was also suitable for the goat species.

The aim of this study was to compare the seminal quality of ejaculates of goats of the Murciano Granadina (MG) breed after freezing and thawing using three extenders. One of them was based on egg yolk and had been successfully used in freezing porcine semen [30], another was based on skimmed milk supplemented with glucose [31,32], and finally, a new extender was designed by our team, also with egg yolk, in which membrane-protective surfactants are replaced with sodium dodecyl sulfate (SDS), which we have called IMIDA. The aim is to find out if they provide different post-thaw semen qualities and to determine which would be the best extender to use in the freezing of semen in the caprine species.

## 2. Materials and Methods

### 2.1. Ethics

Through the experiments, animals were handled carefully, avoiding any unnecessary stress. All experiments were performed following relevant guidelines and regulations. The study was carried out in compliance with the ARRIVE guidelines (https://arriveguidelines.org/ accessed on the 20 January 2021).

### 2.2. Origin of the Animals

The animals used in this study belong to our research team. Our team owns a small experimental farm to carry out our research, and all the males used were born in our facilities.

### 2.3. Animals and Study Design

For this study, we used ejaculates from six bucks of the Murciano Granadina breed aged between 7 and 9 months. All males were housed in individual pens of 10 m^2^, with a covered area and an outdoor exercise yard. The bucks were fed taking into account their nutritional needs. The starting weight of the animals at 7 months was 45 kg. The feeding program was adjusted to the body weight: It started with 50 g of commercial concentrate and ended with 850 g. The commercial concentrate contained 15.2% crude protein, 3.70% crude fat, 16.5% crude fiber, and mineral elements Ca, P, Na, and Mg. Bucks were given ad libitum mineral stones with Zn, 2850 mg/stone; Mn, 1.800 mg/stone; I, 400 mg/stone; Fe, 200 mg/stone; Co, 50 mg/stone; and Se, 55 mg/stone as trace elements and Na, 36%; Ca, 2.5%; and Mg, 0.50%, as macroelements. Each male always had water ad libitum.

Semen was extracted weekly from each male using an artificial vagina, with an immobilized female as a sexual stimulus. The volume of each ejaculate was measured using a graduated glass tube, after which they were diluted 1:10 with Krebs Ringer Phosphate (KRP) medium [31] tempered at 37 °C. A droplet of 30 μL of this dilution was observed under an optical microscope (100×) (Leica Mycrosystem^®^ DM 2000, Barcelona, Spain) to determine the individual motility (IM, where 0 means that they are immobile and 5 that have a very fast progressive movement) [33] and the percentage of fresh mobile spermatozoa (% MOT). The ejaculate with an IM of 3.5 or more and a % MOT of at least 60% was accepted for freezing. The dilutions of the ejaculates of the six study bucks were mixed to avoid differences in initial qualities distorting the final results. The mixture was divided into three parts, and each part was frozen using one of the three extenders (Thilmant, Skimmed Milk and IMIDA). This process was repeated for 8 weeks. We also wanted to check the difference in seminal quality individually for each male, so individual samples were also frozen for an additional 6 weeks. We froze samples from males 1, 2, and 3 with the three extenders. These three males were selected for their high libido and for providing ejaculates in each week of the study. We extracted three ejaculates from each male in each session, which were mixed together (no mixing of different males).

Eight pools were obtained from all males (8 pools × 3 extenders = 24 samples) and with individual samples from three males obtained in 6 weeks (6 weeks × 3 individual males × 3 extenders = 54 samples) for evaluation after thawing.

The composition of three extenders (fraction A) is as follows (Table 1, Table 2 and Table 3).

The three extenders contain, for 1 L of dilution, the following quantity of antibiotics: 0.657 g of streptomycin, 0.657 g of penicillin, 0.15 g of lincomycin, and 0.3 g of spectinomycin (Sigma-Aldrich, Steinheim, Germany). The composition of fraction B was the same as fraction A + 8% of glycerol (Sigma-Aldrich, Steinheim, Germany).

### 2.4. Sperm Processing

Semen with KRP was allowed to cool to room temperature (23 °C), and once tempered, it was centrifuged at 800× *g* for 20 min. The supernatants were removed, and each pellet was resuspended with the calculated amount of extender A (a fraction of the extender without glycerol) to obtain a final straw concentration of 200 × 10^6^ sperm. Each sample was refrigerated so that the temperature dropped to 5 °C within 2 h. Once this temperature was reached, the second part of the corresponding extender, extender B, was added, this time containing glycerol to 8% (its final concentration will be 4%). This addition of the extender plus the glycerol was performed five times at 5 min intervals. It was left to equilibrate for 90 min, during which time the samples were packed into 0.5 mL straws. After 90 min of calibration, the straws were placed in the cryo-freezer (Computer Controlled Rated-Freezer 14 S, Sy-Lab Cube) associated with a computer program to ensure the temperature was reduced gradually. The freezing ramp is as follows [23]: start: T = 5 °C. Cooling rate: −1 °C/minute to −4.5 °C (fusion and seeding temperature). Hold for 1 min at −4.5 °C. Freezing −30 °C/min down to −180 °C. They were then submerged in liquid nitrogen and stored in tanks until they were thawed.

Thawing was done by immersing the straw in a bath with water at 56 °C for 12 s [30]. It was paper-dried, one end was cut off, and the contents placed in a tube with 1 mL of KRP extender at 37 °C. All samples were kept in KRP medium tempered at 37 °C for the hours of analysis.

### 2.5. Sperm Analysis

A straw from each sample of each freezing date was thawed for analysis using the CASA automated system (Computer Assisted Sperm Analysis System, ISAS^®^, PROISER SL, Valencia, Spain). The ejaculations of males 1, 2, and 3 and a mixture of ejaculates of the six males (pool) were analyzed with this system. The samples were diluted 1:3 in KRP buffer, and a drop of this dilution was placed on a slide heated to 37 °C and analyzed in a Leica^®^ DM1000 phase contrast microscope at 100× (Barcelona, Spain). The parameters evaluated were the number of total mobile sperm (TM), the number of progressive mobile sperm (PM) expressed both as percentage of total mobile sperm and percentage of progressive mobile sperm, the curvilinear velocity (VCL, µm/s), straight line velocity (VSL, µm/s), linearity index (LIN), straightness index (STR), and amplitude of lateral head displacement (ALH). An average of six fields were evaluated per sample [34]. Each sample was evaluated freshly thawed: 2 h after thawing and 5 h after thawing.

A different straw from each sample of the same freezing date was thawed in the same way for subjective microscopic assessment. A 30 μL droplet of the seminal dilution was taken and observed at 100× under an optical microscope (Leica Mycrosystems^®^ DM 2000). Each sample was assigned an IM value between 0 and 5 and an approximate percentage of mobile sperm in relation to the total observed. Another 30 μL droplet was mixed with the same amount of eosin-nigrosin, and a smear was prepared on a slide. It was allowed to dry, and this stain was evaluated at 400× under an optical microscope (Leica Mycrosystems^®^ DM 2000) to determine the percentage of spermatozoa considered to be alive or dead. Those observed in the preparation with white color, therefore presenting membrane integrity, were classified as alive, whilst those spermatozoa with pink color, whose membrane was unstructured, were classified as dead. Two hundred sperm were counted from each sample evaluated for vital stain (VS).

To determine the percentage of intact acrosomes (NAR), 100 μL of the semen–KRP dilution was mixed with 400 μL of acrosome solution [35]. A 30 μL droplet of this solution was observed in a phase contrast microscope (B-350 Optika Microscopes, Italy) at 1000× to determine by direct observation the acrosome integrity. One hundred sperm cells were counted per sample evaluated.

These assessments were performed at H0, H2, and H5 after thawing as well as samples analyzed by CASA.

### 2.6. Statistical Analysis

The average results were subjected to statistical analysis using Statgraphics^®^ Centurion XVI.II software. The averages of each parameter were analyzed using the F-test in the ANOVA table to determine if there are significant differences between the means. Multiple range tests determined which means were significantly different from others. The method currently used to discriminate between means is Fisher’s minimum significant difference (LSD) procedure, with a 95% confidence level (*p* < 0.05).

## 3. Results

The semen quality (IM and % MOT) of the fresh samples of each male is shown in Table 4. After being subjected to the freezing and thawing process, this quality decreased to a greater or lesser extent depending on the used extender and the considered male.

Through the evaluation with the CASA system, for the ejaculate pool, the IMIDA extender provided the highest averages in all parameters evaluated at hour 0 (H0) and hour 2 (H2). At hour 5 (H5), its averages were not always the highest, but there were no significant differences with the Thilmant extender. Only the ALH parameter gave more variable results, being higher than Thilmant in H0 and than skimmed milk in H2. It should be noted that no ejaculate from the pool reached H5 with the skimmed milk extender (Table 5).

Looking at the response of the three test males separately, the skimmed milk extender gave the lowest number of samples reaching H5 (Table 6, Table 7 and Table 8). It worked particularly poorly with male 2, from which no ejaculates with motile sperm in H2 could be obtained frozen with this extender. In the case of the LIN and STR indexes, the averages show that the skimmed milk extender performs better than the other parameters, obtaining significantly higher values in some cases. However, the resistance over time was lower with this extender, and these averages may have been influenced by the fact that the number of samples that could be evaluated was too small for a reliable statistical test. The Thilmant and IMIDA extenders gave very similar results to each other, which seems logical because of the similarity in composition. Even though they were very similar, the IMIDA extender showed the highest averages overall.

As for the VCL values (µm/s), these are higher with the IMIDA extender, as at time 0, they are above 110 µm/s in all males, which corresponds to fast spermatozoa. Contrary to the value of the other parameters, the skimmed milk extender showed particularly high LIN and STR averages, especially the latter, and was sometimes significantly higher than the other extenders. It was also observed that after 5 h of thawing, STR rates were higher than in samples at time 0 after thawing in all males, which might suggest that movement straightness improves after an adaptation time after thawing.

Five hours after thawing, many samples frozen with the skimmed milk extender were found to no longer contain motile spermatozoa. Specifically, data could be obtained neither from male 2 samples frozen with this extender in H2 or H5, nor from male 1, nor from the pool of samples in H5 with the skimmed milk extender. Male 3 did not provide any ejaculate reaching H5 with motile sperm with the Thilmant extender. However, the frozen samples with the IMIDA extender provided results at all hours of evaluation and all males.

Comparison of extenders by subjective microscopic observation (Table 9, Table 10, Table 11 and Table 12) revealed less significant differences than the CASA evaluation system. However, it is interesting to mention that the skim milk extender is also the one that offered the lowest averages throughout this evaluation, with significant differences especially in the freshly thawed samples (H0). The averages for the Thilmant and IMIDA extenders were more similar.

The male pool did not show significant differences in any parameter evaluated at any of the evaluation times, but the trend of lower averages for samples frozen with the skim milk extender was repeated (Table 9).

Male 1 showed significant differences in the samples frozen with skim milk, whose averages were lower than those frozen with the other two extenders in the study (Table 10).

Male 2 repeated what was mentioned in the section on evaluation by CASA regarding the poor resistance over time of his samples when frozen with the extender of skim milk. Only one sample of the six evaluated reached H2 and H5. The extenders Thilmant and IMIDA offered results without significant differences for this male (Table 11).

As for male 3, the lowest averages were also yielded when the skim milk extender was used, with significantly lower VS and % NAR parameters with this extender; however, the resistance over time was lower with the Thilmant extender, as no ejaculate was obtained from this male and extender at H5 (Table 12).

After two hours of thawing, the Thilmant and IMIDA extenders generally showed higher averages than the skim milk extender and provided a greater number of ejaculates.

Five hours after thawing, the extender IMIDA maintained motile sperm in a greater number of samples than the other two extenders, with higher averages in some cases. In those that were outperformed by the extender Thilmant, the differences were not significant. The IMIDA extender provided another advantage that was not initially expected. After microscopic evaluation of the samples, we found that the sperm were seen with greater clarity than when using the Thilmant extender, visualizing the sperm cells individually, which facilitates their evaluation, especially for scoring IM and % MOT.

## 4. Discussion

Numerous factors influence sperm quality after thawing, such as the extender used, the dilution and freezing protocol, or the evaluation procedure [36]. Variability between males is also important, as there are males whose sperm freeze very well and others that produce almost total necrospermia after thawing [8,37]. It should be noted that in this experiment, differences between males in the freezing process were not studied. The aim was to study the response to freezing with different extender, so the statistical analysis focused on looking for significant differences between extenders and not between males, whether it was done with a mixture of ejaculates from different males or by freezing samples from different males individually.

For AI to be successful with frozen semen, there must therefore be an appropriate freezing protocol that uses an extender that provides nutrients and protects the sperm cells, thereby allowing them to survive.

The design of a new extender, IMIDA, was based on the properties of SDS as a surfactant product that stabilizes membrane phospholipids, protecting them against freeze–thaw damage [23,24]. This SDS is contained in the Orvus es Paste of the Thilmant protocol, but our hypothesis was that, added independently, it would provide the same seminal quality at a more economical price.

In our work, we simultaneously evaluated the post-thaw quality of semen from different bucks of the Murciano Granadina breed with similar ages after processing their semen by cryopreservation with three different extenders. We observed notable differences in the results both between the qualities shown by the different males and within the same male against different extenders. We were able to see that the skimmed milk extender was generally the one that produced the worst results, especially in the parameters of TM and PM, and that the resistance over time was lower than with the other two extenders. Other authors such as Carpio-Chuchuca (2015) [38] also found significant differences in cattle when using an extender based on milk or egg yolk in TM, with the percentage of mobile sperm decreasing from 80% to 72.5% and the percentage of live sperm after thawing decreasing from 82.5% with egg yolk to 77.5% with skimmed milk. López (2005) [39] found differences in PM in goat semen, with a decrease from 43.7% for the extender containing 2% egg yolk to 35.25% for the skimmed milk extender.

When looking at the values provided by the CASA system, these are mostly higher than those found in the literature. Thus, Tabarez (2014) [8] found in 1-year-old goats an average TM of 39.6% and PM of only 14.2% after using an extender with 15% fresh egg yolk in freezing. Hernández-Corredor (2014) [36] achieved as best results 30.4% TM and 12.2% PM. Our results showed that the best TM mean was 65.6% in male 1 frozen with the IMIDA extender, and the PM mean for this male and extender was 43.8% at H0. When a mixture of ejaculates (pool) was frozen, 71.8% of TM was achieved with the IMIDA extender, and a PM percentage of 56.1% was achieved with this extender at H0. With the rest of the extenders, the averages ranged from 18.3–56.7% TM and from 10.4–36.7% PM at H0. These averages decreased over time, oscillating at H5 between the values of 13.7–62.3% for TM and 6.0–41.6% for PM. Note that not all ejaculates reached H5 with mobile sperm.

Bravo et al. (2011) [40] published that sperm are mobile if their VCL speed is greater than 20 μm/s and classified sperm as slow, medium, or fast according to whether their VCL was less than 45 μm/s, between 45–75 μm/s, or greater than 75 μm/s (for ram sperm). For the experiment of the present paper, these same values were used; however, the speed of the slow spermatozoa was modified from 20 to 10 μm/s (Tomás, 2007) [41].

Muiño et al. (2006) [42], for bovine ejaculates, established higher velocities, considering sperm slow with a VCL from 20–60 μm/s, mean between 60 and 110 μm/s, and fast exceeding 110 μm/s.

With both of the two assessment methods, our samples showed VCL values greater than 110 μm/s in all males when the IMIDA extender was used in freshly thawed samples. Only male 2 did not reach this speed with the skimmed milk extender, and for the pool and for male 3, the VCL was lower with the Thilmant extender. In all cases, the VCL exceeded 100 μm/s on average. As for its resistance over time, we only had ejaculations of all the males that had mobile spermatozoa at H5 with the IMIDA extender. The skimmed milk extender obtained the lowest mean VCL at H5 and the lowest number of ejaculates that resisted until evaluation. Time resistance is of great importance because ejaculates with very good averages but that do not resist over time may be unable to fertilize the oocyte if it is not yet ready to be fertilized at the time of AI. However, if the sperm remain viable, non-reactive, and progressively motile for 1 to 5 h after insemination, the chances of fertilization of the egg increase.

High VCL averages indicate highly active sperm, but they do not have to be progressive. Linearity (LIN) and straightness (STR) indexes are used to determine the progression. LIN reflects the straightness of the sperm trajectory and should be greater than 59% [41]. STR reflects the straightness of movement. For a sperm to be considered progressive, its STR must exceed 80% [43]. Our average results for these indices are well above the minimum values of 59% and 80% at H0 (the only exception is male 3 with the IMIDA extender, which had 78.3% STR). These good averages are maintained over time and are even higher in many of the samples that last until H2 and H5, so it seems that, after a period of acclimatization after thawing, the sperm movement begins to be more rectilinear than in the newly thawed sperm.

Using the traditional method of subjective evaluation, Valencia et al. (1994) [44] reported sperm motility values greater than 59% after thawing. Vallecillo et al. (2004) [45] evaluated the individual motility of cryopreserved semen with Triladyl with 20% egg yolk, obtaining a value of 60.5%, and Luzardo (2010) [46] obtained 38.6% and 35.3% of mobile sperm with Tris yolk and Triladyl, respectively. Konyali et al. (2013) [47] evaluated IM using the Tris extender at two egg yolk concentrations (2% and 20%) and one based on skimmed milk, obtaining values of 58.2%, 70.1%, and 69.7%, respectively. Our results for the percentage of mobile spermatozoa in freshly thawed samples ranged from 57.5% to 74.1% for the IMIDA extender, 55.6% to 65.8% for the Thilmant extender, and 37.5% to 65.0% for the skimmed milk extender depending on the male. These results are slightly superior to those found by Vázquez et al. (2001) [1], who, in the same breed, reported an average of 3.6 in IM, 49.7 % MOT, 55.3% alive with VS, and 68.7% NAR.

These results show that, indeed, the IMIDA extender containing SDS provides similar or even superior post-thaw semen quality averages to the Thilmant extender. The addition of this anionic detergent has been shown to be beneficial for post-thaw sperm quality in several species, such as swine [30], canines [28,29], cattle [24], goats [48], and even felines such as cats [49], used together with an extender containing 20% egg yolk. In goats, the tests carried out in this study show that it also provides high semen quality after thawing. Peña and Forsberg (2000) [25] found that this beneficial effect occurred if Equex was added immediately prior to freezing, whereas its addition over a longer period, such as during the calibration time, worsened the results. They suggested that prolonged exposure to SDS would confer excessive membrane fluidity, which could be counterproductive. Axnér et al. (2004) [49] published that Equex had a positive effect on the percentage of intact acrosomes immediately after thawing but negatively affected the time resistance of cat spermatozoa maintained at 38 °C after thawing. In the tests performed in this study, prolonged exposure was not found to be counterproductive or to affect time resistance, and on the contrary, the extender containing the SDS was the one that maintained the highest number of samples with viable spermatozoa during all the hours of evaluation. These differences could be due to the species used or the concentration of detergent used. In goats, Aboagla and Terada (2004) [48] also found this beneficial effect on total and progressive motility of frozen spermatozoa using concentrations between 0.035–0.1% SDS. The concentration used in this study was quite higher, namely 1% SDS in the IMIDA extender, and no detriment to sperm quality was found.

Hori et al. (2006) [50] studied in canine spermatozoa whether there were differences when using SDS alone or contained in Orvus es Paste and concluded that both had the same protective effect, using them in the appropriate concentration for that species. Our results also found no significant differences in the majority of the averages using SDS alone (IMIDA extender) compared to the one containing it as Orvus es Paste (Thilmant extender).

Our results show that, indeed, the IMIDA extender containing this SDS provides similar or even better post-thawing semen quality averages than the Thilmant extender, and its cost is lower than that of the Orvus es Paste. It also allows a better visualization of the sperm cells, facilitating their subjective evaluation for use in artificial insemination, which is an interesting and useful added advantage.

## 5. Conclusions

Taking into account the parameters presented at thawing, thawed sperm from egg yolk extenders showed higher values in the vital parameters studied, and in addition, the extender called IMIDA showed higher values in the sperm resistance tests, with the sperm diluted in it showing mobility that lasted more than five hours in the thermal resistance tests. Therefore, we can conclude that according to the semen quality averages presented in vitro, the IMIDA extender and the semen-freezing protocol proposed by Thilmant (1997) are suitable for semen cryopreservation in goats.

## Figures and Tables

**Table 1 animals-13-00309-t001:** Composition of the Thilmant (1997) sperm-freezing extender (fraction A) for a total volume of 1 L (distilled H_2_O + egg yolk).

Fructose (Sigma-Aldrich, Steinheim, Germany)	56.82 g
Sodium bicarbonate (Na HCO^3−^) (Sigma-Aldrich, Steinheim, Germany)	1.0 g
Cysteine (L-Cysteine C_3_H_7_NO_2_S) (Sigma-Aldrich, Steinheim, Germany)	0.099 g
Egg yolk (farm fresh eggs)	227.27 mL
Distilled water (Milli-Q^®^ Direct)	772.7 mL
Equex STM^®^ (Orvus es Paste) (Minitub Ibérica, Tarragona, Spain)	11.29 g

**Table 2 animals-13-00309-t002:** Composition of sperm-freezing extender for skimmed milk (fraction A) for a total volume of 1 L.

Skimmed milk (commercial milk)	1 L
Anhydrous glucose (PRS Panreac, Barcelona, Spain)	1.76 g

**Table 3 animals-13-00309-t003:** Composition of IMIDA sperm-freezing extender (fraction A) for a total volume of 1 L (distilled H_2_O + egg yolk).

Fructose (Sigma-Aldrich, Steinheim, Germany)	56.82 g
Sodium bicarbonate (Na HCO^3−^) (Sigma-Aldrich, Steinheim, Germany)	1.0 g
Cysteine (L-Cysteine C_3_H_7_NO_2_S) (Sigma-Aldrich, Steinheim, Germany)	0.099 g
Egg yolk (farm fresh eggs)	227.27 mL
Distilled water (Milli-Q^®^ Direct)	772.7 mL
Sodium dodecyl sulfate (C_12_H_25_NaO_4_S) (Sigma-Aldrich, Steinheim, Germany)	6 g diluted in 10 mL of distilled water (*)

* Note: Mix until a homogeneous paste is formed. Weigh 11.29 g of this paste and add it to the rest of the ingredients.

**Table 4 animals-13-00309-t004:** Semen quality averages for each male in fresh semen.

Male	IM	% MOT
1 (n = 16)	4.5 ± 0.1	81.4 ± 3.1
2 (n = 16)	4.3 ± 0.1	71.1 ± 3.1
3 (n = 11)	4.0 ± 0.1	63.2 ± 3.7
4 (n = 7)	4.1 ± 0.15	63.2 ± 4.6
5 (n = 7)	4.3 ± 0.15	62.5 ± 4.6
6 (n = 6)	4.3 ± 0.1	74.0 ± 3.5
POOL (n = 8)	3.92 ± 0.23	64.17 ± 7.10

Mean ± standard error. IM, individual motility; % MOT, percentage of motile spermatozoa; n, number of samples.

**Table 5 animals-13-00309-t005:** Average semen parameters obtained by CASA system for the pool of MG goats in resistance test, according to extender (if there is no number: no motile spermatozoa reached that evaluation time with that extender).

POOL	H0			H2			H5		
	THILM	MILK	IMIDA	THILM	MILK	IMIDA	THILM	MILK	IMIDA
% TM	48.0 ^a^ ± 4.7 (n = 8)	42.5 ^a^ ± 4.5 (n = 8)	71.8 ^b^ ± 4.2 (n = 8)	53.0 ^a^ ± 4.1 (n = 8)	43.8 ^a^ ± 3.8 (n = 4)	68.3 ^b^ ± 3.9 (n = 8)	29.9 ± 2.28 (n = 5)	-	35.2 ± 6.85 (n = 5)
% PM	31.3 ^a^ ± 3.8	25.7 ^a^ ± 3.7	56.1 ^b^ ± 3.4	36.8 ^a^ ± 2.3	14.4 ^b^ ± 2.2	54.7 ^c^ ± 2.3	18.6 ^a^ ± 2.36	-	10.2 ^b^ ± 2.58
VCL (µm/s)	101.1 ^a^ ± 3.4	117.2 ^b^ ± 3.3	138. ^c^ ± 3.0	118.0 ^a^ ± 3.1	96.6 ^b^ ± 2.9	137.5 ^c^ ± 2.9	88.8 ± 5.68	-	72.0 ± 6.83
VSL (µm/s)	74.5 ^a^ ± 3.0	91.1 ^b^ ± 2.9	121.3 ^c^ ± 2.7	100.8 ^a^ ± 4.1	78.5 ^b^ ± 3.8	126.9 ^c^ ± 4.0	79.8 ± 7.13	-	61.4 ± 4.75
LIN	63.5 ^a^ ± 1.4	72.9 ^b^ ± 1.4	82.9 ^c^ ± 1.3	78.3 ^a^ ± 2.0	78.3 ^a^ ± 1.8	88.5 ^b^ ± 1.9	81.7 ± 2.4	-	82.0 ± 2.9
STR	77.6 ^a^ ± 1.2	81.7 ^b^ ± 1.1	89.5 ^c^ ± 1.0	86.3 ^a^ ± 1.2	91.0 ^b^ ± 1.1	92.8 ^b^ ± 1.2	89.9 ^a^ ± 1.5	-	95.8 ^b^ ± 1.8
ALH	2.6 ^a^ ± 0.06	2.2 ^b^ ± 0.06	2.4 ^c^ ± 0.05	2.2 ^a^ ± 0.06	2.4 ^b^ ± 0.06	2.2 ^a^ ± 0.06	1.84 ± 0.09	-	1.77 ± 0.1

Mean ± standard error. Different letters in the same row for the same time indicate significant differences between extenders. Equal n (number of samples) for all parameters. % TM, percentage of total mobile sperm; % PM, percentage of progressive mobile sperm; VCL, curvilinear velocity; VSL, straight line velocity; LIN, linearity index; STR, straightness index; ALH, amplitude of lateral head displacement. *p* < 0.05.

**Table 6 animals-13-00309-t006:** Average semen parameters obtained by CASA system for male 1 in the resistance test, according to extender (if there is no number: no motile spermatozoa reached that evaluation time with that extender).

MALE 1	H0			H2			H5		
	THILM.	MILK	IMIDA	THILM.	MILK	IMIDA	THILM.	MILK	IMIDA
% TM	55.7 ^a^ ± 3.9 (n = 6)	49.8 ^a^ ± 4.2 (n = 6)	65.6 ^b^ ± 2.8 (n = 6)	39.6 ^a^ ± 5.2 (n = 6)	18.1 ^b^ ± 5.6 (n = 4)	43.3 ^a^ ± 4.1 (n = 6)	43.8 ^a^ ± 5.8 (n = 3)	-	62.3 ^b^ ± 3.2 (n = 4)
% PM	35.0 ^a^ ± 3.1	36.7 ^b^ ± 3.4	43.8 ^b^ ± 2.3	25.8 ^a^ ± 4.2	8.9 ^b^ ± 4.5	26.3 ^a^ ± 3.3	37.9 ± 4.6	-	41.6 ± 2.5
VCL (µm/s)	120.9 ^a^ ± 5.0	129.4 ^a^ ± 5.5	142.7 ^b^ ± 3.7	113.3 ± 6.4	108.9 ± 6.9	107.4 ± 5.0	133.3 ^a^ ± 4.72	-	122.1 ^b^ ± 2.59
VSL (µm/s)	99.4 ^a^ ± 5.8	119.5 ^b^ ± 6.4	114.3 ^b^ ± 4.3	98.2 ± 7.1	87.8 ± 7.6	85.2 ± 5.6	129.5 ^a^ ± 5.4	-	108.7 ^b^ ± 3.0
LIN	75.6 ^a^ ± 1.8	88.6 ^b^ ± 1.9	74.6 ^a^ ± 1.3	79.0 ± 2.6	73.8 ± 2.8	73.3 ± 2.1	92.7 ± 3.6	-	86.4± 2.0
STR	85.8 ^a^ ± 1.08	93.5 ^b^ ± 1.2	85.8 ^a^ ± 0.8	87.5 ± 1.5	90.7 ± 1.6	91.0 ± 1.1	96.3 ± 3.7	-	94.3± 2.0
ALH	2.3 ^a^ ± 0.09	2.1 ^b^ ± 0.08	2.8 ^c^ ± 0.06	2.1 ^a^ ± 0.06	2.7 ^b^ ± 0.07	2.6 ^b^ ± 0.05	1.8 ^a^ ± 0.1	-	2.3 ^b^ ± 0.05

Mean ± standard error. Different letters in the same row for the same time indicate significant differences between extenders. Equal n (number of samples) for all parameters. % TM, percentage of total mobile sperm; % PM, percentage of progressive mobile sperm; VCL, curvilinear velocity; VSL, straight line velocity; LIN, linearity index; STR, straightness index; ALH, amplitude of lateral head displacement. *p* < 0.05.

**Table 7 animals-13-00309-t007:** Average semen parameters obtained by CASA system for male 2 in resistance test, according to extender (if there is no number: no motile spermatozoa reached that evaluation time with that extender).

MALE 2	H0			H2			H5		
	THILM.	MILK	IMIDA	THILM.	MILK	IMIDA	THILM.	MILK	IMIDA
% TM	56.7 ^a^ ± 4.2 (n = 6)	18.2 ^b^ ± 4.5 (n = 6)	54.6 ^a^ ± 3.2 (n = 6)	40.1 ± 5.0 (n = 5)	-	49.1 ± 4.8 (n = 4)	51.6 ^a^ ± 4.6 (n = 3)	-	53.2 ^b^ ± 8.6 (n = 3)
% PM	33.9 ^a^ ± 3.4	10.4 ^b^ ± 3.7	34.1 ^a^ ± 2.6	20.6 ^a^ ± 4.4	-	40.1 ^b^ ± 4.3	27.2 ± 2.1	-	34.2 ± 6.3
VCL (µm/s)	120.6 ^a^ ± 4.6	102.7 ^b^ ± 5.0	119.3 ^a^ ± 3.5	112.6 ± 9.6	-	127.3 ± 8.7	92.8 ± 11.1		100.5 ± 8.2
VSL (µm/s)	93.5 ^a^ ± 5.4	77.0 ^b^ ± 5.8	97.5 ^a^ ± 4.1	102.6 ± 9.6	-	107.3 ± 9.3	83.7 ± 5.6	-	83.5 ± 11.0
LIN	72.8 ^a,b^ ± 2.7	68.7 ^b^ ± 2.9	72.8 ^a,b^ ± 2.7	84.7 ± 4.1	-	75.2 ± 4.0	82.5 ± 5.2	-	76.2 ± 3.8
STR	87.7 ^a^ ± 1.9	80.9 ^b^ ± 2.1	87.7 ^a^ ± 1.5	92.7 ± 2.5	-	88.4 ± 2.4	88.8 ^a^ ± 1.4	-	93.9 ^b^ ± 1.0
ALH	2.8 ^a^ ± 0.1	2.2 ^b^ ± 0.1	2.4 ^c^ ± 0.08	2.0 ^a^ ± 0.07	-	2.4 ^b^ ± 0.07	1.8 ^a^ ± 0.1	-	2.3 ^b^ ± 0.08

Mean ± standard error. Different letters in the same row for the same time indicate significant differences between extenders. Equal n (number of samples) for all parameters. % TM, percentage of total mobile sperm; % PM, percentage of progressive mobile sperm; VCL, curvilinear velocity; VSL, straight line velocity; LIN, linearity index; STR, straightness index; ALH, amplitude of lateral head displacement. *p* < 0.05.

**Table 8 animals-13-00309-t008:** Average semen parameters obtained by CASA system for male 3 in the resistance test, according to extender (if there is no number: no motile spermatozoa reached that evaluation time with that extender).

MALE 3	H0			H2			H5		
	THILM.	MILK	IMIDA	THILM.	MILK	IMIDA	THILM.	MILK	IMIDA
% TM	30.6 ^a^ ± 4.4 (n = 6)	44.1 ^b^ ± 7.3 (n = 6)	54.7 ^b^ ± 3.9 (n = 6)	32.4 ^a^ ± 3.2 (n = 5)	36.4 ^a^ ± 4.6 (n = 4)	58.7 ^b^ ± 3.5 (n = 5)	-	22.6 ^a^ ± 2.8 (n = 2)	33.0 ^b^ ± 2.1 (n = 2)
% PM	20.7 ^a^ ± 3.5	30.4 ^b^ ± 5.9	36.6 ^b^ ± 3.1	17.4 ^a^ ± 2.9	9.1 ^a^ ± 4.1	37.9 ^b^ ± 3.1	-	9.2 ± 3.1	16.2± 2.3
VCL (µm/s)	107.1 ^a^ ± 6.7	112.6 ^a,b^ ± 11.3	133.7 ^b^ ± 5.9	77.7 ^a^ ± 4.21	67.7 ^a^ ± 6.0	124.0 ^b^ ± 4.5	-	86.9 ± 10.18	99.6 ± 7.7
VSL (µm/s)	84.8 ± 6.1	98.2 ± 10.2	97.7 ± 5.4	65.6 ^a^ ± 4.6	58.7 ^a^ ± 6.6	99.9 ^b^ ± 5.0	-	77.9 ± 12.4	83.4 ± 9.4
LIN	71.0 ^a^ ± 2.5	83.4 ^b^ ± 4.2	69.4 ^a^ ± 2.2	75.1 ± 2.1	80.8 ± 3.0	75.8 ± 2.3	-	83.0 ± 5.0	76.4 ± 3.8
STR	80.8 ^a^ ± 1.9	91.5 ^b^ ± 3.2	78.3 ^a^ ± 1.7	86.0 ^a^ ± 1.5	92.8 ^b^ ± 2.2	83.4 ^a^ ± 1.6	-	92.5 ± 2.4	91.9 ± 1.8
ALH	2.3 ^a^ ± 0.1	2.0 ^a^ ± 0.2	2.7 ^b^ ± 0.1	1.9 ^a^ ± 0.06	1.7 ^a^ ± 0.09	2.4 ^b^ ± 0.07	-	2.0 ± 0.16	2.2 ± 0.12

Mean ± standard error. Different letters in the same row for the same time indicate significant differences between extenders. Equal n (number of samples) for all parameters. % TM, percentage of total mobile sperm; % PM, percentage of progressive mobile sperm; VCL, curvilinear velocity; VSL, straight line velocity; LIN, linearity index; STR, straightness index; ALH, amplitude of lateral head displacement. *p* < 0.05.

**Table 9 animals-13-00309-t009:** Average semen quality parameters by subjective microscopic evaluation for the pool of males in endurance test, according to extender.

POOL	H0			H2			H5		
	THILM.	MILK	IMIDA	THILM.	MILK	IMIDA	THILM.	MILK	IMIDA
IM	3.9 ± 0.4 (n = 8)	4.0 ± 0.4 (n = 8)	4.1 ± 0.4 (n = 8)	3.75 ± 0.35 (n = 8)	3.75 ± 0.35 (n = 4)	4.0 ± 0.35 (n = 8)	** 3 (n = 2)	** 3 (n = 2)	** 3 (n = 3)
% MOT	67.5 ± 3.0	62.5 ± 2.97	66.2 ± 3.0	53.7 ± 4.89	57.5 ± 4.89	63.7 ± 4.89	27.5 ± 1.98	18.7± 1.98	20.0 ± 2.79
VS (% alive)	56.0 ± 6.3	47.0 ± 6.3	61.5 ± 6.3	52.5 ± 8.01	42.0 ± 8.01	49.5 ± 8.01	38.5 ± 6.49	32.0 ± 6.49	45.0 ± 9.18
% NAR	78.0 ± 4.5	74.0 ± 4.5	72.5 ± 4.5	78.5 ± 4.92	67.2 ± 4.92	67.5 ± 4.92	69.5 ± 5.71	61.0 ± 5.71	59.0 ± 8.08

Mean ± standard error. Equal n (number of samples) for all parameters. IM, individual motility; % MOT, percentage of motile spermatozoa; VS, vital stain; % NAR, percentage non-reacted acrosomes. *p* < 0.05. ** no standard error because all measurements were equal.

**Table 10 animals-13-00309-t010:** Average semen quality parameters by subjective microscopic evaluation for male 1 in endurance test, according to extender.

MALE 1	H0			H2			H5		
	THILM.	MILK	IMIDA	THILM.	MILK	IMIDA	THILM.	MILK	IMIDA
IM	3.9 ^a^ ± 0.2 (n = 6)	3.8 ^a^ ± 0.1 (n = 6)	4.4 ^b^ ± 0.1 (n = 6)	3.9 ± 0.22 (n = 6)	3.8 ± 0.22 (n = 4)	4.4 ± 0.15 (n = 6)	3.6 ± 0.48 (n = 3)	3.3 ± 0.39 (n = 3)	4.0 ± 0.25 (n = 4)
% MOT	65.0 ^a^ ± 8.1	37.5 ^b^ ± 8.1	74.1 ^a^ ± 5.8	66.2 ^a^ ± 8.44	48.1 ± 8.44	65.6 ± 5.97	60.0 ± 13.0	25.0 ± 10.62	48.6 ± 6.95
VS (% alive)	71.7 ^a^ ± 9.1	39.4 ^b^ ± 8.2	67.8 ^a^ ± 6.5	64.5 ^a,b^ ± 7.71	43.0 ^a^ ± 7.71	66.2 ^b^ ± 5.83	67.0 ^a,b^ ± 13.18	34.3 ^b^ ± 10.76	48.7 ^a^ ± 7.05
% NAR	75.8 ^a^ ± 6.6	60.2 ^b^ ± 5.9	76.2 ^a^ ± 4.7	74.0 ^a^ ± 6.94	57.2 ^b^ ± 6.94	75.9 ^a^ ± 5.24	72.5 ± 12.44	51.7 ± 10.15	53.6 ± 6.65

Mean ± standard error. Different letters in the same row for the same parameter indicate significant differences between extenders. Equal n (number of samples) for all parameters. IM, individual motility; % MOT, percentage of motile spermatozoa; VS, vital stain; % NAR, percentage non-reacted acrosomes. *p* < 0.05.

**Table 11 animals-13-00309-t011:** Average semen quality parameters by subjective microscopic evaluation for male 2 in endurance test, according to extender.

MALE 2	H0			H2			H5		
	THILM.	MILK	IMIDA	THILM.	MILK	IMIDA	THILM.	MILK	IMIDA
IM	4.0 ± 0.2 (n = 6)	3.7 ± 0.3 (n = 6)	4.1 ± 0.2 (n = 6)	4.0 ± 0.26 (n = 5)	* 4.0 (n = 1)	4.4 ± 0.16 (n = 4)	3.5 ± 0.22 (n = 2)	* 3.75 (n = 1)	4.2 ± 0.18 (n = 2)
% MOT	55.6 ± 12.3	65.0 ± 18.1	62.7 ± 9.0	43.3 ^a,b^ ± 10.25	* 37.5 ^a^	65.6 ^b^ ± 6.28	10.0 ^a^ ± 3.63	* 20.0 ^a^	60.8 ^b^ ± 2.96
VS (% alive)	61.9 ± 7.1	53.0 ± 10.0	63.6 ± 5.0	62.7 ^a^ ± 7.81	* 32.0 ^b^	55.5 ^a^ ± 5.12	47.5 ^a^ ± 5.59	* 12.0 ^b^	53.2 ^a^ ± 4.57
% NAR	74.5 ^a^ ± 7.6	56.0 ^b^ ± 10.8	65.8 ^a^ ± 5.34	81.7 ^a^ ± 10.71	* 45.0 ^b^	63.9 ^a,b^ ± 7.57	82.0 ^a^ ± 1.88	* 62.0 ^b^	70.9 ^b^ ± 1.54

Mean ± standard error. Different letters in the same row for the same parameter indicate significant differences between extenders. Equal n (number of samples) for all parameters. IM, individual motility; % MOT, percentage of motile spermatozoa; VS, vital stain; % NAR, percentage non-reacted acrosomes. *p* < 0.05. * only 1 measurement.

**Table 12 animals-13-00309-t012:** Average semen quality parameters by subjective microscopic evaluation for male 3 in endurance test, according to extender (if there is no number: no motile spermatozoa reached that evaluation time with that extender).

MALE 3	H0			H2			H5		
	THILM.	MILK	IMIDA	THILM.	MILK	IMIDA	THILM.	MILK	IMIDA
IM	4.1 ± 0.2 (n = 6)	4.1 ± 0.2 (n = 6)	3.95 ± 0.1 (n = 6)	3.8 ± 0.14 (n = 5)	3.9 ± 0.17 (n = 4)	3.7 ± 0.12 (n = 5)	-	** 3.75 ^a^ (n = 2)	** 3.5 ^b^ (n = 2)
% MOT	65.8 ± 9.3	58.7 ± 11.4	57.5 ± 7.2	46.7 ± 15.0	47.5 ± 18.4	45.6 ± 13.0	-	38.7 ^a^ ± 0.88	50.0 ^b^ ± 0.88
VS (% alive)	68.3 ^a^ ± 8.0	28.5 ^b^ ± 9.8	50.2 ^a^ ± 6.2	63.7 ^a^ ± 9.8	21.0 ^b^ ± 16.98	55.5 ^a,b^ ± 12.0	-	17.5 ^a^ ± 6.70	58.5 ^b^ ± 6.70
% NAR	76.2 ^a^ ± 11.7	42.5 ^b^ ± 14.3	50.7 ^b^ ± 9.0	76.0 ^a^ ± 10.56	33.0 ^b^ ± 18.3	61.0 ^a^ ± 12.93	-	40.5 ^a^ ± 3.65	68.5 ^b^ ± 3.65

Mean ± standard error. Different letters in the same row for the same parameter indicate significant differences between extenders. Equal n (number of samples) for all parameters. IM, individual motility; % MOT, percentage of motile spermatozoa; VS, vital stain; % NAR, percentage non-reacted acrosomes. *p* < 0.05. ** no standard error because all measurements were equal.

## Data Availability

The original contributions presented in the study are included in the article, and further inquiries can be directed to the corresponding author.

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
