# Peer review of "Post-Thaw Quality of Spermatozoa Frozen with Three Different Extenders in the Murciano Granadina Goat Breed"

_animals, 2023, doi:10.3390/ani13020309_

Round 1

Reviewer 1 Report

The subject of the article is interesting and very practical since it compares the effectiveness of three extenders for freezing/thawed semen in the Murciano Granadina goat breed. The final conclusion, supported by sperm motility results obtained by optical microscopy and through the CASA system, indicates that the IMIDA extender (method developed by the work team itself) is the best of the three tested to preserve the quality of thawed semen in time and lower cost than Thilmant thinner.

The introduction is correct, as well as the material and the method, but in the part of the results the legends of the tables should be improved. The tables must be readable without the text, and, for this reason, I think they must contain the full name of the abbreviations used (tables 5 to 8), as well as the meaning of the N in tables 4 to 8. It should, as well, refer to the level of significance considered. Discussion is appropriate, but perhaps this paragraph: “In addition, after microscopic evaluation of the samples, we found that they were seen more clearly than when Thilmant extender was used, visualizing individual spermatozoa, which facilitates their evaluation, especially for MI score and %MOT” should be in the results before discussing it in the text. The conclusions are appropriate to the results obtained.

Reviewer 2 Report

The present article compares the potential of a new self-made freezing extender with common used media for goat semen. The scientific relevance and the applied methods of the study are limited. Parts of the methods are not sufficiently described. Overall, it is questionable whether the results shown here are sufficient for publication. It is advisable to extend the range of methods to support the results with further data on sperm quality.

-       Under which circumstances was the semen stored 5 h after thawing (temperature, oxygen supply, volume)?

-       Was the semen incubated again before motility testing?

-       Table 2 and Table 3: Why are the first ingredients of the thinners written in bold?

-       Table 4:

o   The term macho is uncommon and should be replaced by male or buck, for example.

o   The table description does not contain all the necessary information to be understandable in itself. Please describe the abbreviations IM and MOT exactly.

-       Table 5 - Table 8:

o   The separation of the examination time points is not clear. Please insert at least 2 additional dividing lines

o   Please explain in the table headings why some extender have no test results or which exclusion criterion was used

-       Only advanced CASA parameters are discussed. This is due to the fact that the method spectrum of the investigations is very limited. Further parameters such as the membrane status or functional parameters such as the membrane fluidity of the samples would be advisable here

Round 2

Reviewer 2 Report

The comments were adequately answered and the results were supplemented with classical microscopic examinations. However, the range of methods used in the study remains limited. For further investigations, I strongly recommend to include further flow cytometric data.